

# A minimal set of internal control genes for gene expression studies in head and neck squamous cell carcinoma

Vinayak Palve[1], Manisha Pareek[1], Neeraja M. Krishnan[1],
Gangotri Siddappa[2], Amritha Suresh[2], Moni A. Kuriakose[2]
and Binay Panda[1]

[1] Ganit Labs, Bio-IT Centre, Institute of Bioinformatics and Applied Biotechnology, Bangalore, India
[2] Head and Neck Oncology, Mazumdar Shaw Centre for Translational Research, Bangalore, India

## ABSTRACT

Selection of the right reference gene(s) is crucial in the analysis and interpretation of gene expression data. The aim of the present study was to discover and validate a minimal set of internal control genes in head and neck tumor studies. We analyzed data from multiple sources (in house whole-genome gene expression microarrays, previously published quantitative real-time PCR (qPCR) data and RNA-seq data from TCGA) to come up with a list of 18 genes (discovery set) that had the lowest variance, a high level of expression across tumors, and their matched normal samples. The genes in the discovery set were ranked using four different algorithms (BestKeeper, geNorm, NormFinder, and comparative delta Ct) and a web-based comparative tool, RefFinder, for their stability and variance in expression across tissues. Finally, we validated their expression using qPCR in an additional set of tumor:matched normal samples that resulted in five genes (*RPL30*, *RPL27*, *PSMC5*, *MTCH1*, and *OAZ1*), out of which *RPL30* and *RPL27* were most stable and were abundantly expressed across the tissues. Our data suggest that *RPL30* or *RPL27* in combination with either *PSMC5* or *MTCH1* or *OAZ1* can be used as a minimal set of control genes in head and neck tumor gene expression studies.

# INTRODUCTION

Internal control genes or housekeeping genes are important in obtaining reliable and reproducible data from gene expression studies in cancer (*Eisenberg & Levanon, 2013*). Internal control genes should be abundantly and uniformly expressed across tumor and normal tissues and across different experimental conditions (*Janssens et al., 2004*). Cancers of head and neck region is the sixth most common cancer worldwide with an incidence of 550,000 cases annually (*Ferlay et al., 2010*). Past studies argue the lack of uniformity of expression on internal control genes based on experimental conditions (*De Jonge et al., 2007*; *Greer et al., 2010*). Although there are previous reports that describe the reference or internal control genes in head and neck squamous cell carcinoma

Corresponding author
Binay Panda, binay@ganitlabs.in

(HNSCC) gene expression studies (*Lallemant et al., 2009*), use of high-throughput data like microarrays and RNA-seq from tumors and their matched normal tissues following a confirmation of expression in independent set of samples are scant in the literature. Additionally, it is crucial that the expression of internal control genes remains unaltered across temporal, spatial and experimental conditions that takes into account of the genes with a wide dynamic range of expression. Therefore, revisiting the validity of widely used genes like *ACTB*, *TUBB*, and *GAPDH* is required along with discovery of a minimal set of internal control genes in HNSCC gene expression studies that use expression data from patient cohorts from different geography (*Krishnan et al., 2015*, *2016*) and large consortia like TCGA (*The Cancer Genome Atlas Network, 2015*).

In the present study, we analyzed HNSCC gene expression data from three sources; *in house* microarray data (*Krishnan et al., 2015*, *2016*) ($n = 21$), TCGA RNA-seq data (*The Cancer Genome Atlas Network, 2015*) ($n = 42$) and quantitative real-time PCR (qPCR) data on individual genes from the previously published studies (*Bär, Bär & Lehmann, 2009*; *Eisenberg & Levanon, 2013*; *Lallemant et al., 2009*; *Martin, 2016*; *Vandesompele et al., 2002*); to come up with a set of genes (discovery set) that are stably and robustly expressed with least variance across tumor:normal pairs. We subsequently validated the expression of the discovery set in additional tumor:normal pairs ($n = 14$) using qPCR and obtained a minimal set of two housekeeping genes by comparing and ranking their expression during validation.

# MATERIALS AND METHODS

## Microarray data processing

The gene expression profiling using Illumina HumanHT-12 v4 expression BeadChips (Illumina, San Diego, CA, USA), raw data collection and data processing using the R package lumi (*Du, Kibbe & Lin, 2008*) and individual batch-correction using ComBat (*Johnson, Li & Rabinovic, 2007*) is described previously (*Krishnan et al., 2015*, *2016*). The raw signal intensities from arrays were exported from GenomeStudio for pre-processing and analyzed using *R*. Gene-wise expression intensities for tumor and matched control samples from GenomeStudio were transformed and normalized using variance stabilizing transformation (VST) and LOESS methods, respectively, using lumi (*Du, Kibbe & Lin, 2008*) and top genes with least across-sample variance in expression were selected. Genes with fold change between 0.95 and 1.05 (tumor/normal) and with standard deviation < 0.05 for the fold changes were selected further from a larger gene list at a high level in both the microarray and the RNA-seq data (Table S1).

## RNA-seq data processing

RSEM-processed RNA-Seq gene expression values (TCGA pipeline, Level 3) were downloaded from the old TCGA repository (https://genome-cancer.ucsc.edu/proj/site/hgHeatmap/). The transcripts per kilobase million (TPM) values for all the genes were extracted from the Level 3 files. Further, genes that showed zero or NA values in any of these samples were eliminated, and the log fold change values between the respective tumor and normal samples calculated by taking a log transformation of the ratios between expression values

for tumor and normal. We filtered expression data that fulfilled three criteria for all samples; expressed at $\geq$ 3 TPM, tumor/normal ratio: 0.9–1.1 and standard deviation across all the tissues < 0.5. Finally genes were short-listed from a larger gene list expressed at a high level in both the microarray and the RNA-seq data (Table S1) for the discovery set.

### Discovery set

In addition to the microarray and RNA-seq gene expression data, the most commonly used internal control genes form the HNSCC gene expression literature (*Bär, Bär & Lehmann, 2009*; *Eisenberg & Levanon, 2013*; *Lallemant et al., 2009*; *Martin, 2016*; *Vandesompele et al., 2002*) were also considered to select the discovery set ($n$ = 18) (Table S2). These genes were used for validation in an independent set of tumor:normal pairs ($n$ = 14) using qPCR.

### Patient samples

Patients were accumulated voluntarily after obtaining informed consent from each patient and as per the guidelines from the institutional ethics committee of the Mazumadar-Shaw Cancer Centre (IRB Approval number: NHH/MEC-CL/2014/197). All the tissues were frozen immediately in liquid nitrogen and stored at −80 °C until further use. Only those tumors with squamous cell carcinoma (with at least 70% tumor cells and with confirmed diagnosis) and their adjacent normal tissues as confirmed by histology (Fig. S1) were included in the current study. Patients underwent staging according to AJCC criteria, and curative intent treatment as per NCCN guidelines involving surgery with or without post-operative adjuvant radiation or chemo-radiation at the Mazumdar Shaw Cancer Centre were accrued for the validation study.

### RNA extraction and cDNA synthesis

The total RNA was extracted from 25 mg of tissues using the RNeasy mini kit (cat:74104; Qiagen, Germantown, MD, USA) with on-column digestion of DNA using RNase free DNase set (cat:79254; Qiagen, Germantown, MD, USA ) as per manufacturer's instructions. The RNA quality was checked using gel electrophoresis and Agilent Bioanalyzer. A total of 500 nanogram of total RNA was subjected to cDNA synthesis using Takara's Prime Script first strand cDNA synthesis kit (cat: 6110A).

### Quantitative real-time PCR

Quantitative real-time PCR was carried out using KAPA Biosystem's SYBR Fast qPCR universal master mix (cat: KK4601). The primer sequences for all the 18 discovery set reference genes and the amplification conditions are mentioned in Table 1A. The primers were either designed for this study or were chosen from the literature (*Campos et al., 2009*) or from the online resources (https://primerdepot.nci.nih.gov/ and https://pga.mgh.harvard.edu/primerbank/). All amplification reactions were carried out in triplicates using three negative controls: no template control with nuclease free water (cat: AM9932; Ambion, Austin, TX, USA), no amplification control and no primer control in each amplification plate. We have followed the guidelines for qPCR experiments as suggested previously (*Bustin et al., 2010*).

**Table 1 Primers used in the current study (A).**

A

| No. | Gene | F/R | Sequence | Genomic position | Exon-intron junction | Intron spanning | Amplicon size (bp) | Reference |
|---|---|---|---|---|---|---|---|---|
| 1 | RPL27 | F | ACAATCACCTAATGCCCACA | chr17:43000081-43002884 | √ | √ | 146 | This study |
| | | R | GCCTGTCTTGTATCTCTCTTCAA | | | | | |
| 2 | RPS29 | F | GCACTGCTGAGAGCAAGATG | chr14:49577820-49586363 | √ | √ | 213 | *De Jonge et al. (2007)* |
| | | R | ATAGGCAGTGCCAAGGAAGA | | | | | |
| 3 | OAZ1 | F | GGATCCTCAATAGCCACTGC | chr19:2269621-2271410 | – | √ | 150 | *De Jonge et al. (2007)* |
| | | R | TACAGCAGTGGAGGGAGACC | | | | | |
| 4 | RPL30 | F | ACAGCATGCGGAAAATACTAC | chr8:98041745-98042696 | – | √ | 158 | *De Jonge et al. (2007)* |
| | | R | AAAGGAAAATTTTGCAGGTTT | | | | | |
| 5 | LDHA | F | TTGTTGGGGTTGGTGCTGTT | chr11:18396916-18399514 | – | √ | 137 | This study |
| | | R | AAGGCTGCCATGTTGGAGAT | | | | | |
| 6 | ACTB | F | CCTTGCACATGCCGGAG | chr7:5529603-5530574 | – | √ | 112 | https://primerdepot.nci.nih.gov |
| | | R | GCACAGAGCCTCGCCTT | | | | | |
| 7 | RPS13 | F | CGTCCCCACTTGGTTGAAGTT | chr11:17075584-17077438 | √ | √ | 129 | This study |
| | | R | CGTACTTGTGCAACACCATGT | | | | | |
| 8 | GAPDH | F | GCATCCTGGGCTACACTGA | chr12:6537873-6538138 | – | √ | 162 | *Campos et al. (2009)* |
| | | R | CCACCACCCTGTTGCTGTA | | | | | |
| 9 | TUBA1B | F | GTCGCCTTCGCCTCCTAATC | chr12:49129610-49131330 | – | √ | 146 | This study |
| | | R | TCACTTGGCATCTGGCCATC | | | | | |
| 10 | PSMC5 | F | TTGACGGACCAGAGCAGATG | chr17:63827498-63829528 | √ | √ | 124 | This study |
| | | R | CTCCGGAGGTTTTGGCTCTT | | | | | |
| 11 | MTCH1 | F | CTTGGCGTAGGTGAAGAAGC | chr6:36978607-36985885 | √ | √ | 123 | https://primerdepot.nci.nih.gov |
| | | R | CATCCCCTGCTCTACGTGAA | | | | | |
| 12 | MKRN1 | F | GCAGCAAGGGATGACTTTGT | chr7:140459862-140471905 | – | √ | 98 | https://primerdepot.nci.nih.gov |
| | | R | TGTATTTATGGAGACCGCTGC | | | | | |
| 13 | ERGIC3 | F | TCTCATGCTGCTACTGTTCCT | chr20:35542342-35542833 | √ | √ | 152 | https://pga.mgh.harvard.edu/primerbank/ |
| | | R | CAATACTCAGATAGGCACAAGGC | | | | | |
| 14 | ADRM1 | F | CAATGCTCCTCATCCTGGTC | chr20:62304519-62306228 | – | √ | 91 | https://primerdepot.nci.nih.gov |
| | | R | GGAGGGTCTACGTGCTGAAG | | | | | |
| 15 | RPL37A | F | TTCTGATGGCGGACTTTACC | chr2:216499955-216501378 | – | √ | 115 | https://primerdepot.nci.nih.gov |
| | | R | ATGAAGAGACGAGCTGTGGG | | | | | |
| 16 | RPL5 | F | TCGTATAGCAGCATGAGCTTTC | chr1:92837537-92840589 | – | √ | 136 | https://primerdepot.nci.nih.gov |
| | | R | TGTTGCAGATTACATGCGCT | | | | | |
| 17 | TSPAN31 | F | GGCTATTAACCGAAGCAAACAGA | chr12:57746235-57746683 | – | √ | 117 | https://pga.mgh.harvard.edu/primerbank/ |
| | | R | GTGAGGTTGAATAAGCCACAACA | | | | | |
| 18 | DARS | F | AGCCGCAAGAGTCAGGAGA | chr2:135979349-135985450 | – | √ | 124 | https://pga.mgh.harvard.edu/primerbank/ |
| | | R | CCCGAACCAAAACTCGATCTG | | | | | |

**B**

**Ranking of candidate reference genes**

| Rank | Genorm | | Normfinder 1 | | Comparative Ct | | Bestkeeper | | Reffinder comprehensive | |
|---|---|---|---|---|---|---|---|---|---|---|
| | Genes | Stability value (M) | Genes | Stability value (M) | Genes | $2^{-\Delta\Delta Ct}$ | Genes | Pearson correlation coefficient | Genes | Geomean ranking |
| 1 | RPL30 | 0.254 | PSMC5 | 0.004 | RPL27 | 1.13 | RPL27 | 0.278 | RPL30 | 1.32 |
| 2 | RPL27 | 0.254 | MTCH1 | 0.005 | RPS13 | 0.90 | RPL30 | 0.321 | RPL27 | 1.57 |
| 3 | PSMC5 | 0.348 | TSPAN21 | 0.006 | RPL30 | 1.21 | RPL5 | 0.362 | PSMC5 | 3.35 |
| 4 | OAZ1 | 0.365 | DARS | 0.006 | RPL5 | 0.82 | RPS13 | 0.391 | OAZ1 | 5.03 |
| 5 | MKRN1 | 0.375 | RPL30 | 0.007 | RPL37A | 0.76 | MTCH1 | 0.419 | MTCH1 | 5.38 |

**Note:**
Results from different tools on the internal control genes and their ranks as obtained using the validation samples (B).

## Statistical analysis

For stability comparison of the internal candidate reference genes obtained using the discovery set, we analyzed the validation data independently using four most commonly used algorithms, Genorm (*Vandesompele et al., 2002*), Normfinder (*Andersen, Jensen & Ørntoft, 2004*), Bestkeeper (*Pfaffl et al., 2004*), and Comparative Ct (*Schmittgen & Livak, 2008*). In addition to the comparative Ct method, we also calculated the values of mean standard deviation (*Silver et al., 2006*) of Ct for each gene. The results from the above tools were compared with an online tool Reffinder (*Xie et al., 2012*) and the results were interpreted and presented separately. Graphpad prism software version 5 was used to analyze the data and plot the graphs.

## RESULTS AND DISCUSSION

The schema for selecting genes for discovery set is depicted in Fig. 1. After analyzing data from all sources, 18 genes were selected that had least variance across all the tumor and normal samples, were well annotated and had some biology known (Table 1A; Table S2). Standard curves reflecting the linear regression ($R^2 > 0.9$) and amplification efficiency (Fig. S2; Table S3) were generated for the select candidate genes. Primers for all the genes showed specific amplifications as shown by the dissociation curves (Fig. S3) and by gel electrophoresis of the amplified products (Fig. S4). Data from qPCR validation of genes (Fig. 2A; Table S3) showed variable levels of expression for the 18 genes in different validation samples. As Fig. 2B indicates independent results from the four different algorithms (Genorm, Normfinder, Bestkeeper, and Comparative Ct) and the online tool RefFinder. As the tool RefFinder does not take amplification efficiency into account and merely sums up results from other tools, we did not rely on the results from RefFinder but used it as one of the tools along with the other four to interpret the results independently. Out of the 18 genes across all the samples, we ranked the expression of top 11 stably expressed genes namely, *RPL30, RPL27, PSMC5, OAZ1, MTCH1, TSPAN21, DARS, MKRN1, RPS13, RPL5,* and *RPL37A*. Although, there was some

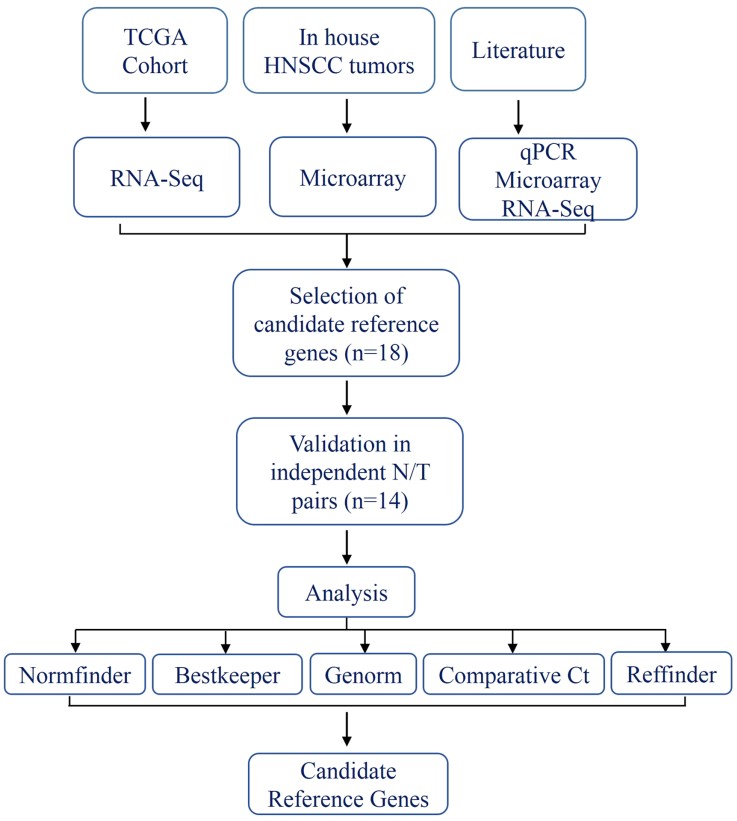

**Figure 1 Schema indicating the selection criteria and validation of the internal control genes.**

inconsistency between the top-ranked genes among the different algorithms/tools, for the genes *RPL27* and *RPL30* the data was consistent across all the algorithms. As both of *RPL27* and *RPL30* belong to the same ribosomal family proteins, and in order to avoid any potential error due to mutual expression alterations of those genes, we considered inclusion of the next three genes *PSMC5*, *MTCH1*, and *OAZ1*, the expression of which were stable across all the samples. Therefore, *RPL27* or *RPL30* in combination with any one of the three *(PSMC5/MTCH1/OAZ1)* fulfilled all the three criteria for ideal internal control genes, least variance across samples, high-level of expression in both tumors and normal and at the top of the rank by the algorithms tested.

The aim of the current study was not to select a set of genes for a particular tumor stage and/or differentiation and in response to drug treatment but to select genes that are robustly expressed across multiple tumor stages and differentiation and across tumor and their matched normal samples. In HNSCC, previous attempts have been made to come up with a set of internal control genes using qPCR data (*Lallemant et al., 2009*; *Martin, 2016*) or using tumors and healthy controls (*Reddy et al., 2016*). However, it is important that the discovery is made from a transcriptome-wide study that takes into account data from tumors and their matched control samples. This is especially important as we have data on HNSCC from large consortia, like TCGA (*The Cancer Genome Atlas Network, 2015*), and data from HNSCC from non-TCGA cohorts

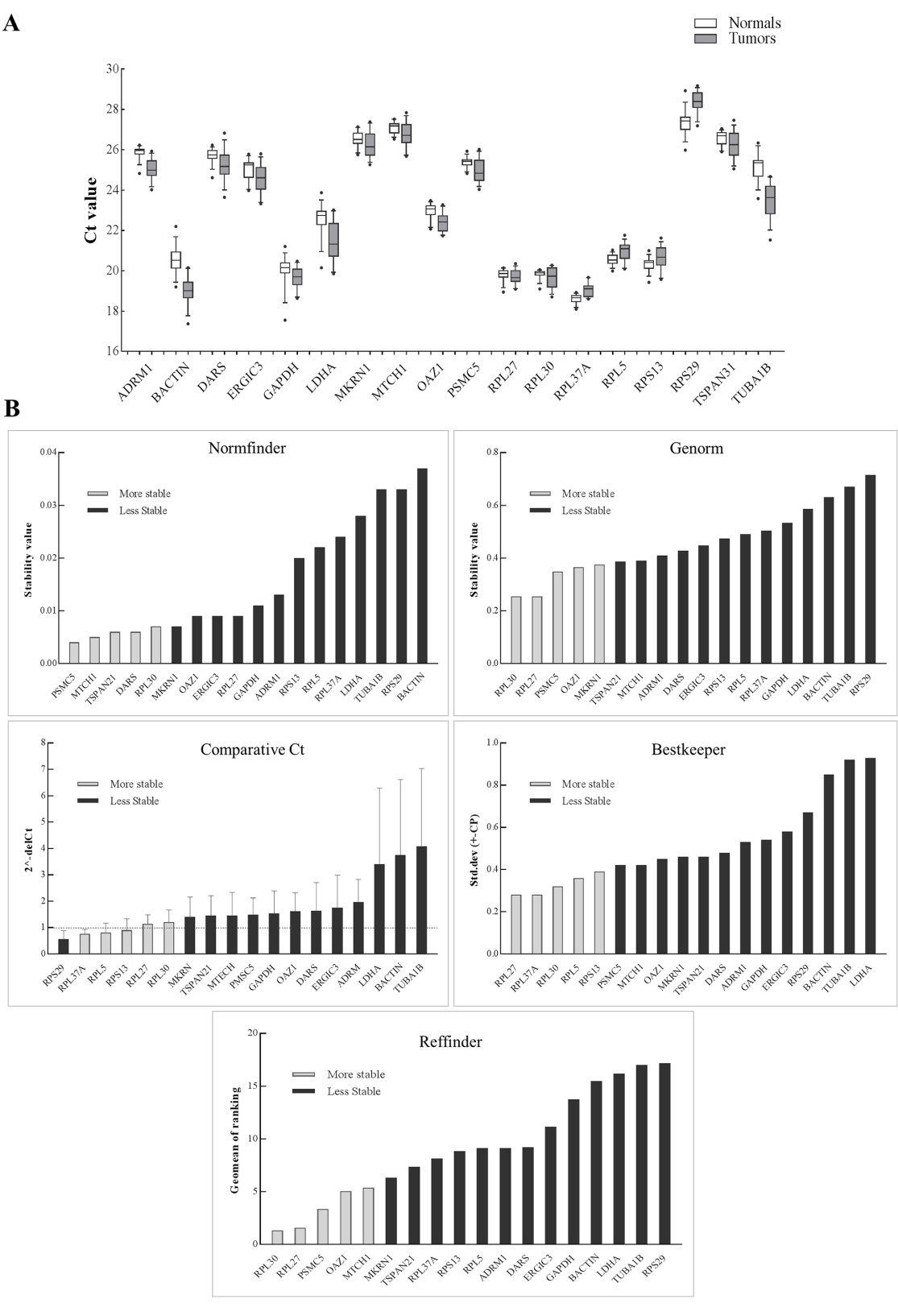

PeerJ ______________________________________________

**Figure 2 Expression of genes in the discovery set across tumors and normal samples.** (A) The Ct values were generated from qPCR experiments and plotted as box plots with median values as lines and boxes indicating 10–90 percentile values and whiskers as the maximum and minimum values. Ranking of 18 genes using different tools (B). *Y*-axis for different tools represents: stability value (Genorm, Normfinder), standard deviation ± (CP) (Bestkeeper), $2^{-(deltaCt)}$ (Comparative Ct method) and geomean of ranking (RefFinder). The columns in light grey represent stable genes across tumors and normal samples.

(*Krishnan et al., 2015, 2016*). It is not surprising that the genes in our discovery set code for structural proteins (ribosomal, cell structure and integrity) and perform essential functions like glycolysis and translation (Table S2). In the validation study, *RPL30*, *RPL27*, *PSMC5*, *OAZ1*, and *MTCH1* came as the top five ranked genes with stable expression across all the samples. Both *RPL27* and *RPL30* were previously found to be one the most stable candidate internal control genes (*De Jonge et al., 2007*; *Lallemant et al., 2009*) with stable expression in salivary samples of oral cancer patients (*Martin, 2016*). Additionally, previous findings on non-HNSCC tumors had *RPL27*, *RPL30*, and *MTCH1* in the list of top internal control genes (*Popovici et al., 2009*). Although, our data supports *PSMC5*, *OAZ1*, and *MTCH1* were stably expressed and ranked after *RPL30* and *RPL27* by RefFinder, they were expressed at a lower level than *RPL30* and *RPL27*. Therefore, we suggest using *RPL30* or *RPL27* in combination with one of the three genes (*PSMC5*, *MTCH1*, and *OAZ1*) to serve as a minimal set of control genes in head and neck tumor gene expression studies.

Our study suffers from some shortcomings. First, our sample size was small. Second, the qPCR studies, although highly sensitive and widely used, could have resulted in a bias in our gene expression data. RNA input volume, cDNA synthesis efficiency, pipetting volumes and accuracy, and primer amplification efficiencies are some of the factors that may affect qPCR results (*Bustin et al., 2009*). Other factors that might have influenced our selection of the discovery set are inter-platform and -assay variability, temporal, spatial, and experimental conditions (for example, drug treatment), and presence of transcript variants in tumor samples. A larger sample size for different experimental conditions and data from cell lines assayed by using a single kit and a single platform (for example, total transcriptome RNA-seq using a single provider's platform and assay version) may provide answers to these in the future. Third, the selection of genes in our study might have gotten biased as a result of the field effect (due to use of adjacent normal tissues). Use of true normal tissues may circumvent this issue. However, due to the stringency of the ethics committee rules to obtain additional normal tissues from patients without a clinical reason(s), studies with true normal tissues, outside of the adjacent normal, are least likely to take place. Fourth, in our study, like previously reported (*Popovici et al., 2009*), a difference in the ranking of genes may be tool-specific and certain parameters (like qPCR amplification efficiency) might have influenced the ranking of the genes. A better design and a robust statistical tool to rank genes may overcome this limitation in the future.

## CONCLUSIONS

Although past studies are heavily reliant on the use of *ACTB*, *TUBB*, and *GAPDH* to normalize expression data, evidence suggests that the expression of these genes vary

greatly and are not suitable internal control genes (*Glare et al., 2002*; *Oliveira et al., 1999*; *Selvey et al., 2001*; *Zhong & Simons, 1999*). Additionally, expression of internal control genes may vary based on experimental conditions and tumor type. Therefore, we used a larger discovery set to validate internal control genes across HNSCC tumors and their paired normal samples to come up with a set of robust and stably expressed genes (*RPL30*, *RPL27*, *PSMC5*, *MTCH1*, and *OAZ1*) across tissues.

## LIST OF ABBREVIATIONS

**HNSCC**    Head and neck squamous cell carcinoma
**NTC**    No template control
**NAC**    No amplification control
**NPC**    No primer control.

## ACKNOWLEDGEMENTS

The authors thank Janani Hariharan for helping with the RNA-seq data analyses.

### Funding

Research presented in this article is funded by Department of Electronics and Information Technology, Government of India (Ref No: 18(4)/2010-E-Infra., 31-03-2010) and Department of IT, BT and ST, Government of Karnataka, India (Ref No: 3451-00-090-2-22). The funders had no role in study design, data collection and analysis, decision to publish, or preparation of the manuscript.

### Grant Disclosures

The following grant information was disclosed by the authors:
Department of Electronics and Information Technology, Government of India: 18(4)/2010-E-Infra.
Department of IT, BT and ST, Government of Karnataka, India: 3451-00-090-2-22.

### Competing Interests

The authors declare that they have no competing interests.

### Author Contributions

- Vinayak Palve conceived and designed the experiments, performed the experiments, analyzed the data, prepared figures and/or tables, authored or reviewed drafts of the paper, approved the final draft.
- Manisha Pareek performed the experiments, approved the final draft.
- Neeraja M. Krishnan analyzed the data, authored or reviewed drafts of the paper, approved the final draft.
- Gangotri Siddappa contributed reagents/materials/analysis tools, approved the final draft.
- Amritha Suresh contributed reagents/materials/analysis tools, approved the final draft.

- Moni A. Kuriakose contributed reagents/materials/analysis tools, approved the final draft.
- Binay Panda conceived and designed the experiments, prepared figures and/or tables, authored or reviewed drafts of the paper, approved the final draft.

## Human Ethics

The following information was supplied relating to ethical approvals (i.e., approving body and any reference numbers):

The Mazumdar-Shaw Cancer Centre Ethics Committee granted approval for the study (IRB Approval number: NHH/MEC-CL/2014/197).

## Data Availability

The raw data are provided in the Supplemental Files.

## Supplemental Information

Supplemental information for this article can be found online at http://dx.doi.org/10.7717/peerj.5207#supplemental-information.

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
