# Peer review of "A minimal set of internal control genes for gene expression studies in head and neck squamous cell carcinoma"

_PeerJ, doi:10.7717/peerj.5207_

## Round 0.1 · original submission · Major Revisions

The article is clearly written and well structured.

However, some aspects have to be clarified/changed to make the manuscript suitable for the publication in PeerJ, as suggested by the reviewers. In particular, it is crucial:

1) To clarify/change the method used for comparing and selecting the most stable reference genes (see reviewer 2);
2) To use Mean Standard Deviation (MeanStdDev) of Ct (see reviewer 3);
3) Select at least one more gene from another metabolic pathway to exclude an error due to a mutual expression alteration of the ribosomal genes (see reviewer 3);
4) Discuss the possible bias of considering tissue adjacent to the tumor as a control (in fact, the immune cells infiltrating normal tissue next tumor may be similar to those infiltrating the tumor, making the tissue "not normal") (see reviewer 1)

Reviewer 1 ·

Basic reporting

Please check at line 115 the term "previously"

Experimental design

The weaknesses of the study are well described from line 169 to line 188.
In particular, the use of the tissue adjacent to the tumor as a control could represent a strong bias; the authors could add a histological characterization of this tissue in order to confirm its non-tumor origin.

Validity of the findings

no comment

Reviewer 2 ·

Basic reporting

the article is clearly written and well structured.

Experimental design

Although the topic of reference gene selection for RT-qPCR experiments is not novel, the experimental set-up by which the authors first screen microarray and sequencing data to come up with candidate reference genes makes this paper stand out.
However, I have one major concern about the method used for comparing and selecting the most stable reference genes. The authors do not sufficiently disclose which software package was used to ompare the reference genes. This is important as some applications (such as Reffinder) do not take the PCR efficiencies into account. And this can impact the ranking of the genes, see De Spiegelaere et al. (Plos One, https://doi.org/10.1371/journal.pone.0122515)

The authors should use the original software packages for Genorm, Normfinder and Bestkeeper or they should use open source packages (such as packages in the R environment) that allow us to validate whether the algorithms are appropriately used.

Second, the Reffinder tool also performs a combined analysis of reference genes, but here again, the methodology is not based on statistically sound data but a mere summing up of four tools, of which the last (the delta CT) is not even a frequently used method to explore reference gene stability. Hence I would not recommend the authors to use this 'combined analysis' do describe their research results.

Hence, I would strongly recommend that the authors step away from the reffinder tool and use peer-evaluated software packages to reperform their analysis. Or, in case they did not use Reffinder as the sole sourse, then they should clearly write this out.

Validity of the findings

The validity of the finding stands and falls with the use of Reffinder, as small differences in gene ranking may occur when PCR efficiencies are not taken into account.

Reviewer 3 ·

Basic reporting

Well written manuscript that is addressing the internal control genes for gene expression studies in head and neck squamous cell carcinoma. However, some point of major and minor criticism should be considered.

Experimental design

This is an original primary research within Aims and Scope of PeerJ. Research question well defined, relevant and meaningful. Presented study performed to a high technical and ethical standard. Methods described with sufficient detail and information to replicate.

Validity of the findings

The study has a limitation related to the small sample size used for validation. However, to select a preliminary list of genes, the authors used data obtained  using different platforms (sequencing, microarrays and qPCR), which significantly reduces the probability of error.

Additional comments

Well written manuscript that is addressing the internal control genes for gene expression studies in head and neck squamous cell carcinoma. However, some point of major and minor criticism should be considered.
Major points
1. In the comparative Ct method the authors used 2^ Ct values (also see the difference of designation in table 1B and Figure 2B), but the classic comparative Ct method operates with values of Mean Standard Deviation (MeanStdDev) of Ct for each genes in pairs with each other genes from author's discovery set (see ref. Silver et al., 2006). Comparative Ct method is similar to geNorm methods but uses only Ct values and do not uses amplification efficiency as geNorm does. Authors should add the MeanStdDev values to the manuscript and change the Results and Discussion section accordingly.
2. Dissociation curves are not enough for evidence of specific amplification. Analysis of PCR products by gel electrophoresis using DNA markers should be done.
3. According to the results provided by the authors the RPL30 and RPL27 genes show the most stable expression indeed. However, choosing the genes as a minimal set of reference genes has an important limitation that both genes belong to the same metabolic pathway. Therefore, the authors should propose at least one more gene from the other metabolic pathway to exclude an error due to a mutual expression alteration of the ribosomal genes.
Minor points
1. On Supplementary Figures 1 and 2 provide the diagrams in the same order for reader convenience.
2. It should be provided the exact values of amplification efficiency for each gene based on the amplification efficiency graphs on Supplementary Figures 1.
3. In Table 1B delete the "1" from "Normfinder 1" column name or explain what does in mean.
4. On Figure 2B all axis Y should be named or none of them.
5. Correct the typo in word "Quantitiatve" in line 106.

References
(Silver et al., 2006) Silver, N., Best, S., Jiang, J., & Thein, S. L. (2006). Selection of housekeeping genes for gene expression studies in human reticulocytes using real-time PCR. BMC molecular biology, 7(1), 33.

---

## Round 0.2 · Minor Revisions

The manuscript is greatly improved. Please add to the manuscript the information suggested by Reviewer 2.

Reviewer 1 ·

Basic reporting

In my opinion, the manuscript is clearly written, well structured and the references are appropriate

Experimental design

The authors have answered all the points discussed by the reviewers, and the experimental design of the manuscript has now been improved

Validity of the findings

Conclusions are well stated, and this manuscript can be very useful for gene expression studies in head and neck squamous cell carcinoma

Additional comments

The authors have answered all the points discussed by the reviewers, in this way the manuscript has now been improved

Reviewer 2 ·

Basic reporting

The article is well structured and clear

Experimental design

The experimental design is good

Validity of the findings

The validity of the findings is supported by the results.

Additional comments

I have no major comments on this article. However, since the authors cite the papers from Bustin et al. on the MIQE guidelines I would urge them to provide all data. This means showing all replicates of the standard curves in supp Fig 2, and providing at least a list of all Cq values or preferably provide the rdml files of the qPCR runs.

Reviewer 3 ·

Basic reporting

Well written manuscript. There are no other comments.

Experimental design

There are no other comments.

Validity of the findings

There are no other comments.

Additional comments

After the corrections performed by authors I have no other comments and I would recommend the manuscript for publication.

---

## Round 0.3 · accepted · Accept

The manuscript is greatly improved and deserves to be published in PeerJ.

#